# Research

behaviour, ecology

*Loxodonta africana*, noise, risk-avoidance, vibrational communication, seismic vibration

**Author for correspondence:**
Beth Mortimer
e-mail: beth.mortimer@zoo.ox.ac.uk

# Noise matters: elephants show risk-avoidance behaviour in response to human-generated seismic cues

Beth Mortimer[1], James A. Walker[1], David S. Lolchuragi[2], Michael Reinwald[1] and David Daballen[2]

[1]Department of Zoology, University of Oxford, Oxford OX1 3PS, UK
[2]Save the Elephants, Marula Manor, Karen, Nairobi 00200, Kenya

BM, 0000-0002-7230-3647; MR, 0000-0001-8481-8392

African elephants (*Loxodonta africana*) use many sensory modes to gather information about their environment, including the detection of seismic, or ground-based, vibrations. Seismic information is known to include elephant-generated signals, but also potentially encompasses biotic cues that are commonly referred to as 'noise'. To investigate seismic information transfer in elephants beyond communication, here we tested the hypothesis that wild elephants detect and discriminate between seismic vibrations that differ in their noise types, whether elephant- or human-generated. We played three types of seismic vibrations to elephants: seismic recordings of elephants (elephant-generated), white noise (human-generated) and a combined track (elephant- and human-generated). We found evidence of both detection of seismic noise and discrimination between the two treatments containing human-generated noise. In particular, we found evidence of retreat behaviour, where seismic tracks with human-generated noise caused elephants to move further away from the trial location. We conclude that seismic noise are cues that contain biologically relevant information for elephants that they can associate with risk. This expands our understanding of how elephants use seismic information, with implications for elephant sensory ecology and conservation management.

## 1. Introduction

African elephants use a variety of sensory modes, whether for communication through signals [1,2] or for information gathering by detecting cues generated by other animals (e.g. predator odour [3]), humans (e.g. voices [4], roads/railways [5]) or natural earth processes (e.g. rainfall [6]). Information transfer through ground-based, or seismic, vibrations is the least well understood [7]. Elephants likely detect seismic information using the Pacinian corpuscles on the feet and/or the inner ear, picking up ground vibrations via bone conduction [8]. Elephants are thought to use seismic signals to communicate with each other, as seismic vibrations are generated by elephants during certain infrasonic vocalizations, known as rumbles [9–11]. The rumbles contain both acoustic and seismic components in the frequency range 20–40 Hz and under [12–14], which are modelled to propagate to a maximum of 6 km under differing favourable conditions [15–17].

Sensory modes can be investigated using playback experiments, including seismic information transfer in elephants. Behavioural changes during these experiments indicate what signals and cues animals can detect and what they can discriminate between, which involves classifying the potential information according to the source identity and status [18]. A series of acoustic (no seismic component) playback experiments with elephants has shown that elephants can detect, for example, elephant rumbles, the sound of bees, human voices and big

**2**

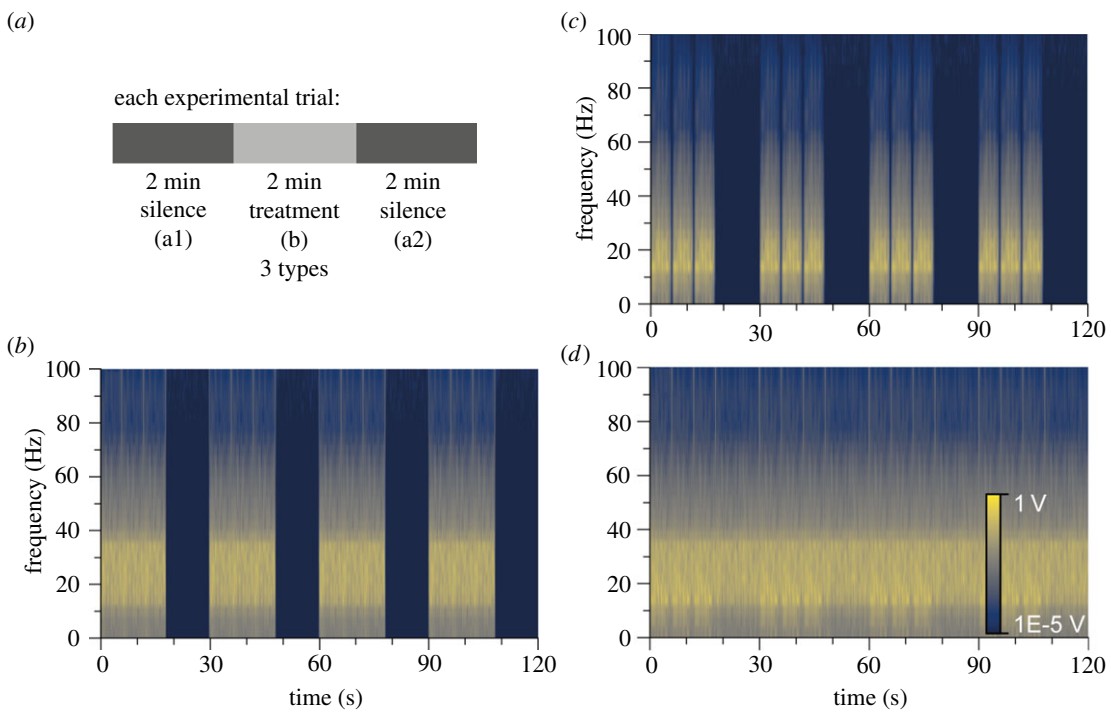

**Figure 1.** Design of playback tracks. (*a*) Two-minute treatments (light grey) were between 2 min of silence (dark grey) for each playback track. Treatments were three types: (*b*) elephant-generated seismic vibrations (Ele); (*c*) white/human-generated noise (WN); (*d*) combined track of WN superimposed on elephant track (Ele&WN). Colour gives spectral amplitude on a log scale from dark blue (1E−5 V) to yellow (1 V). Note that propagation alters the tracks, as seen when a geophone records the seismic vibrations 10 m from the source (electronic supplementary material, figures S1 and S2). (Online version in colour.)

cat growls [4,12,19–21]. Elephants can also discriminate between subtle differences in acoustic sources, changing their behavioural response, for example human voices versus bees [11], or even gender, age and ethnicity of human voices [4]. Two studies to date have used seismic playback experiments to explicitly test elephant responses to ground-based vibrations [22,23]. These showed that seismic playback of (the acoustic component of) an elephant alarm rumble is sufficient to elicit a behavioural response [23] and the elephants responded significantly to seismic playback of alarm rumbles from familiar elephants, but not from unfamiliar elephants or artificial warble tones [22].

Yet potential seismic information for elephants is wider than elephant-generated signals, encompassing what may traditionally be considered as noise, but can be used to inform biological decision making, making them cues [24]. What is generally referred to as seismic noise is generated by a mixture of biotic, abiotic and human sources, so can be natural or artificial (human-generated). These seismic vibrations could potentially provide direct reliable information about the seismic source for decision making in elephants, so are potential seismic cues. Relevant seismic sources include elephants or other animals that will generate seismic vibrations as they move around [13,15,25], the processes of the natural environment (e.g. thunder [14]), and humans that purposefully or incidentally generate seismic vibrations (e.g. wind turbines [26]). Elephants could use these seismic vibrations as cues to determine the presence of potential threats, for example from the movement patterns of other animals or from human activity [4,11].

Aside from potentially acting as cues, the detection of seismic noise is also useful for decision making as higher noise decreases communication efficacy [18]. A lower signal-to-noise ratio (SNR; i.e. higher noise level) reduces the ability to detect and discriminate between seismic signals and cues (when the signal magnitude is constant) [15]. Therefore, detecting and responding to seismic noise can allow elephants to mitigate these negative effects (e.g. by using repeated or louder calls [18]) or choose environments where they are more likely to be able to communicate effectively through seismic vibrations (the seismic domain). However, seismic noise is inevitable as it is superimposed during wave propagation [15]. SNR naturally decreases as propagation distance increases, so elephants must have methods to cope with increasing noise levels since they can respond to acoustic rumbles over a variety of distances, up to the kilometre range [12,27].

Despite the potential sources of seismic information, it remains untested whether and how elephants respond to seismic vibrations beyond elephant-generated signals (their infrasonic rumbles) [22,23]. Here, we focus on whether seismic 'noise' acts as a cue for elephants, testing how wild elephants respond to seismic vibrations of different noise types (elephant- versus human-generated noise). Since there is increasing scope for anthropogenic seismic noise within the elephants' natural and captive landscapes [28,29], determining whether human-generated seismic noise can act as cues for elephants has important implications for their conservation management. This includes understanding the potential impacts on elephants of land-use change that is expected to increase noise, for example due to infrastructure development (roads, railways, wind farms, etc.) [30], which includes seismic noise [26,31].

## 2. Methods

### (a) Seismic playback tracks and calibration

Three 6 min seismic playback tracks of different noise types were generated as stimuli to play to wild elephants in the field (figures 1 and 2*a*; see also electronic supplementary material,

*Proc. R. Soc. B* **288**: 20210774

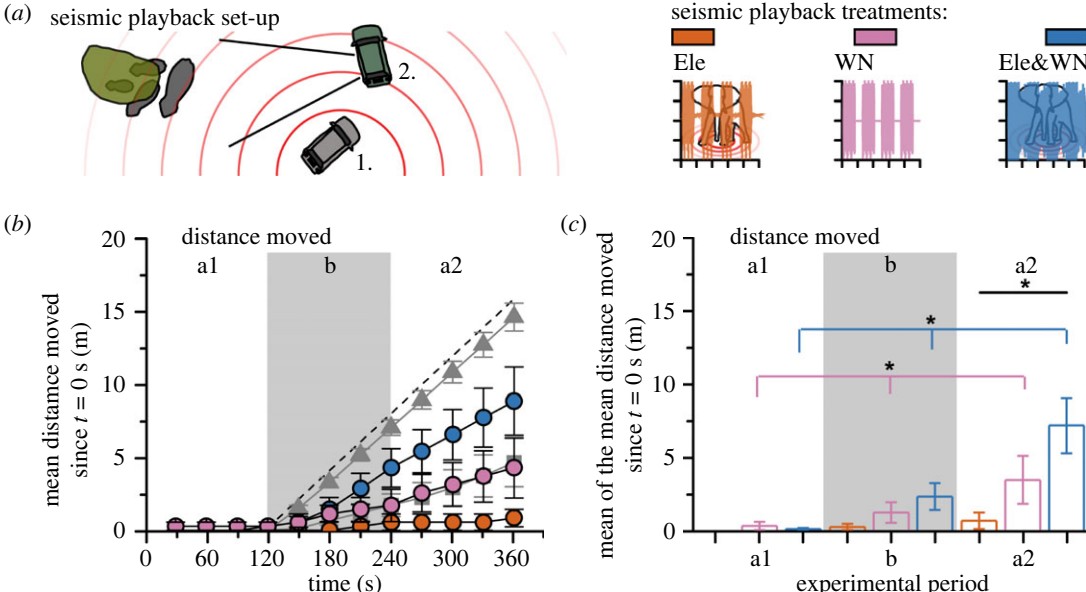

**Figure 2.** Elephant behavioural responses suggest detection and discrimination of seismic treatments that differ in noise type. (*a*) Seismic playback set-up, where car 1 plays the seismic tracks to resting elephants (grey) under trees (green) and car 2 records their behaviour within a camera's field of view (indicated with solid black lines), as well as audio and seismic data. Three seismic playback treatments were used: elephant-generated seismic vibrations (Ele; orange), white/human-generated noise (WN; pink) and a combined track of WN superimposed on the elephant track (Ele&WN; blue), where time-voltage for 2 min is shown (+1 V to −1 V). (*b*) Mean distance moved over time (t) since $t = 0$ s ($n = 7$, families where data were collected across all three seismic treatments). Dashed line gives the maximum distance that could be moved since $t = 120$ s, as capped by the distance-coding method. This method was used to code the data from [32], with acoustic WN (grey squares, $n = 15$) and acoustic bee noise (grey triangles, $n = 17$) played from $t = 120–360$ s. Grey shaded area gives experimental period b, $t = 120–240$ s, when the seismic treatment is applied. (*c*) The same seismic treatment data as B, but plotting the mean across each experimental period. Error bars give standard error of the mean between families. Lines, brackets and asterisks denote significant difference from Friedman's test, where $p < 0.05$. (Online version in colour.)

Methods). The first treatment was a source function [15] of recordings of elephant-generated seismic vibrations (Ele). The Ele source function was 6 s long and was repeated thrice (18 s long). The second seismic playback treatment was 18 s of human-generated white noise (WN; random magnitude over time, with many frequencies at equal intensities) generated in MATLAB. This track was used to test whether the source type of the seismic vibration influenced elephant behaviour, where the frequency range was equivalent (electronic supplementary material, figures S1 and S2). For the final treatment, Ele and WN were added together, by adding amplitudes over every time step, to overlay the WN onto the elephant-generated track (Ele&WN), generating a track of longer exposure to simulate elephant-generated vibrations in a noisy environment.

Twelve seconds of silence (for Ele and WN) or WN (for Ele&WN) were added to the end of the 18 s segments in Audacity (freeware) software. These 30 s segments were repeated four times to generate the 2 min seismic treatment period (b). Two minutes of silence were added either side to create the silent control periods (a1 and a2) to finish the 6 min playback tracks (figure 1).

For the playback, a custom-built and portable system was used to generate the seismic cues using a partially buried modified speaker (see electronic supplementary material, Methods). The playback system was portable to allow it to be deployed in the field at the elephants' chosen resting sites under trees. The tracks were played into the speaker from an iPad at full volume (Apple, USA) via an amplifier (Pyle PLMRA400). Tracks were trialled and recorded using a microphone (Earthworks M30 microphone) and vertical geophone (Raspberry Shake 4D, USA; 100 fps) at 3, 10 and 30 m from the playback location (encompassing playback distances to elephants). This was to ensure that there was a high signal in the seismic domain compared to (i) background seismic levels when tracks were silent and (ii) the acoustic domain (electronic supplementary material, figures S1 and S2).

The amplitudes of tracks and geophone recordings differed—for the input tracks, amplitudes were similar across all three treatments (figures 1 and 2*a*). From our recordings, the mean maximum amplitude of seismic vibrations was lower for 10 s of Ele track compared to WN and Ele&WN 10 m from the speaker ($5E{-}6 \pm 1E{-}6$, $1.2E{-}5 \pm 4E{-}6$ and $1.2E{-}5 \pm 4E{-}6$ m s$^{-1}$, respectively, standard deviation given $n = 4$; electronic supplementary material, figures S1 and S2), which was also seen at 3 m ($7E{-}6 \pm 4E{-}6$, $1.3E{-}5 \pm 2E{-}6$, $1.4E{-}5 \pm 2E{-}6$ m s$^{-1}$) and 30 m ($4E{-}6 \pm$ less than $1E{-}6$, $8E{-}6 \pm 1E{-}6$, $7E{-}6 \pm 2E{-}6$ m s$^{-1}$) from the speaker. These were on average above background seismic noise levels ($4E{-}6 \pm 2E{-}6$, $3E{-}6 \pm 3E{-}6$, $2E{-}6 \pm$ less than $1E{-}6$ at 3, 10 and 30 m, respectively).

## (b) Field experiments

All institutional and national guidelines for the use of protected animals for scientific research were followed during this project. The project was approved through an ethical review process within the University of Oxford by the Zoology Animal Welfare and Ethical Review Board (ref.: APA/1/5/ZOO/NAPSA/Mortimer/ElephantVibrations) and the project was approved locally via a research permit granted from National Commission for Science, Technology and Innovation, Kenya for research in Samburu county (ref.: NACOSTI/P/16/69501/9147) and via an approved research affiliation with the Kenya Wildlife Service. Park permits were gathered for Samburu and Buffalo Springs National Reserves. Animal welfare standards were ensured during fieldwork via ongoing evaluation of elephant behaviours during experiments.

In the field, the seismic tracks were played to wild elephants within the Samburu and Buffalo Springs National Reserves in Kenya (electronic supplementary material, Methods). For each trial, one randomly chosen 6 min track was played to each family/sub-family (table 1). The focal elephant was selected based on their size and visibility. The family or sub-family

**Table 1.** Independent families/sub-families sampled during field trials, indicating the order of three seismic playback treatments (1st, 2nd and 3rd indicated with yellow, red and blue respectively) and group size (1–6).

| family/sub-family name | treatment | | |
| --- | --- | --- | --- |
| | Ele | WN | Ele&WN |
| Artists 1 | 1st: 3 | 3rd: 2 | 2nd: 3 |
| Artists 2 | 3rd: 4 | 1st: 3 | 2nd: 2 |
| Biblical towns | | | 1st: 3 |
| Butterfly 2 | 1st: 2 | | |
| Clouds | 1st: 2 | | |
| First ladies | 2nd: 4 | 1st: 2 | 3rd: 4 |
| Hardwoods | 2nd: 2 | 1st: 3 | 3rd: 5 |
| Hardwoods b | | 1st: 4 | |
| Native Americans | 1st: 3 | 3rd: 2 | 2nd: 3 |
| Royals | 2nd: 4 | 1st: 3 | 3rd: 3 |
| Spices | | 2nd: 4 | 1st: 3 |
| Storms | 1st: 6 | | |
| Swahilis | 1st: 4 | 3rd: 1 | 2nd: 4 |
| Virtues 1 | | 1st: 2 | |
| Virtues 1b | | 2nd: 4 | 1st: 3 |
| Winds 3 | | | 1st: 4 |
| **totals = 16 families, seven with all treatments** | **10** | **11** | **11** |

of the focal elephant was taken to be an independent group of elephants for analysis and were identified both in the field and using photographs.

The speaker was an average of 17.3 m (standard deviation: 3.9 m, $n = 32$) from the position of the focal individual at the start of the trial. From reference to our trial experiments (at 3, 10 and 30 m), this propagation distance gives maximum amplitude levels at *ca* less than $1E{-}5\,\mathrm{m\,s^{-1}}$. For approximate comparison, this is under the range of maximum seismic amplitude recorded from elephant steps and car noise ($2.9E{-}5$ and $4.4E{-}5\,\mathrm{m\,s^{-1}}$; electronic supplementary material, figure S3).

To avoid habituation, each family/sub-family was left a minimum of 5 days before another experimental trial with a different treatment [4,20,22]. Therefore, treatment was only 2 min every 6 days (or more) for each family/sub-family group. Sixteen independent families/sub-families were sampled in total, with seven families receiving all three seismic treatments. Group size during playback (defined as individuals within 10 m of the focal individual that was greater than three quarters the size of the largest elephant) varied from 1 to 6 (table 1).

During each experiment, video, audio and seismic data were recorded and synchronized. Recording equipment was deployed an average of $8.4 \pm 3.6$ m ($n = 32$) from the speaker. The video camera was used to record the behavioural responses of the focal individual (Sony RX II; 30 fps). The geophone and microphone recordings were used as reference data at each field location.

## (c) Analysis of behaviour

To code most behaviours (for distance travelled, see below), each video was cropped into the three 2 min experimental periods (a1, b, a2; figure 1a) and given a random code to eliminate bias. The videos were then analysed blind using BORIS software to extract the durations of behaviours of interest [33], which included vigilant, social, eating and other types of behaviours (electronic supplementary material, table S1). In MATLAB, behaviour durations were converted to time budgets, which was defined as the total duration divided by the time the relevant body part was visible within an experimental period.

Distance moved from trial location (i.e. initial focal elephant position) was calculated from frames 30 s apart, which were compared to code whether the focal elephant had moved more than a body length during that period (adding 2 m for yes, or 0 m for no). The distance moved since the start of the trial ($t = 0$ s) was a cumulative sum over the length of the trial. This allowed a conservative measure of the distance moved from the trial location to be calculated, without estimating the absolute distance moved from the video frames, which would add error. Using this method, there was a maximum (+2 m every 30 s) and a minimum possible distance moved (0 m throughout). The direction of travel was not coded, but for all elephants, distance moved was always away from the speaker, with none coming back towards their initial position (see video data on Dryad data repository). The videos were coded blind by two researchers independently.

This method was also applied to the data of King *et al.* [32], which gives the number of families remaining stationary in response to acoustic WN and the acoustic sound of African bees. The remaining stationary over a 30 s period was coded as +0 m, and not remaining stationary over a 30 s period was coded as +2 m. Again, the direction of travel was not explicitly coded, but elephant responses were away from the trial location [32].

Non-parametric statistics were used to analyse the data, including Friedman's tests, Mann–Whitney U tests and Wilcoxon signed-rank tests, depending on the number of variables being compared and whether there were repeated measures or not (electronic supplementary material, Methods). Data are shown and *p*-values are given where statistical tests indicated a significant difference, defined as $p < 0.05$.

## 3. Results and discussion

Our study supported the hypothesis that elephants are able to detect and discriminate between seismic vibrations of different noise types (figure 2*b,c*). Most notably, we found significant differences in the distance moved by elephants in response to the three seismic treatments (figure 2*b,c*). We found that treatments containing human-generated noise led to elephants significantly increasing the distance that they moved away from the trial location: in both the WN and combined noise treatment (Ele&WN), the elephants significantly increased the distance moved away from the trial location over the experimental periods ($p < 0.01$ for both treatments; Friedman's test $\chi^2(2) = 9.29$ and 10). Furthermore, the elephants did not respond in the same way to the three treatments: the three treatments differed significantly in the distance moved during the post-treatment period ($p = 0.048$; Friedman's test $\chi^2(2) = 6.08$). Over all the trials, 1/10 (Ele), 2/11 (WN) and 5/10 (Ele&WN) focal elephants left the site during or following treatments, with instances of running away from the trial location for the combined treatment (Ele&WN; electronic supplementary material, movie S1). For the families that were exposed to all three treatments ($n = 7$), treatment order did not have a significant effect on distance moved: the distance moved did not significantly differ due to whether it was first, second or third track for any of the treatment periods ($p = 0.607$, 0.466, 0.619;

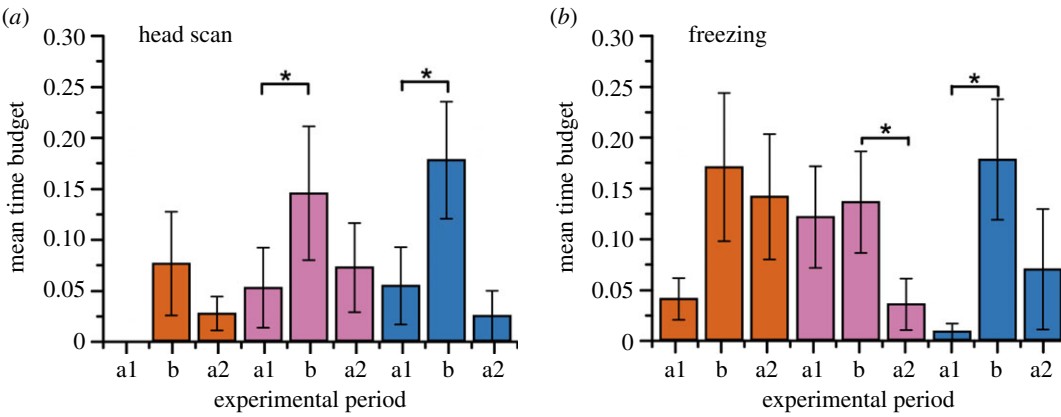

**Figure 3.** Elephant time budget responses show significant differences in behaviour for control and treatment periods. Three seismic playback treatments were used: elephant-generated seismic vibrations (Ele; orange), white/human-generated noise (WN; pink) and a combined track of WN superimposed on elephant track (Ele&WN; blue). (a) Mean time budget of elephants showing head scanning behaviour during each treatment period ($n = 10$ families for Ele and Ele&WN, $n = 11$ for WN): a1 was pretreatment sampling, $t = 0$–120 s, and a2 was post-treatment sampling, $t = 240$–360 s. (b) Mean time budget of elephants showing freezing behaviour during each treatment period (sample size as in A). Error bars give standard error of the mean between families. Brackets and asterisks denote significant paired difference from Wilcoxon signed-rank test between treatment periods, where $p < 0.05$. (Online version in colour.)

Friedman's test $\chi^2(2) = 1$, 1.53, 0.96 for a1, b and a2 periods respectively). This finding differs from previous seismic playback experiments, where warble tone 'noise' (with frequency content and duration similar to an alarm rumble) did not elicit a behavioural response [22]. This indicates that the amplitude and/or time of exposure of seismic noise is important to elephants when assessing its risk, which requires further study in the field.

Elephants have been shown to increase their distance moved in response to the playback of the acoustic sound of bees, human voices, tiger growls and elephant alarm rumbles [4,20–22,32], as well as an increased number of tourist vehicles [34]. To allow a quantitative comparison, we applied our analytical method (see Methods) to the distance moved in response to the sound of bees and acoustic WN, taken from King *et al.* [32] (figure 2*b*). This analysis does not reveal anything about the distance moved by elephants in response to playback treatment beyond the cap of 2 m per 30 s. Interestingly, the distance moved in response to acoustic and seismic WN was similar at 21.0 and 26.0% of the capped maximum and they were not significantly different from each other during any experimental period ($p = 0.172$, $p = 0.639$ and $p = 0.241$; Mann–Whitney $U = 165$, 166.5, 156). Therefore, elephant responses to acoustic and seismic WN for these two studies were comparable. However, elephants were more likely to move further due to the sound of bees than the combined noise seismic track: whereas the response to bees averaged 86.9% of the maximum capped distance moved since the start of treatment (using our coding method; figure 2*b* dashed line; where 100% = +2 m every 30 s, 0% = +0 m), the response to Ele&WN we saw here was 48.1%. Therefore, for the sound of bees, almost every elephant moved at least 2 m every 30 s (13/17 from 150 s onwards), whereas fewer elephants showed this response rate to the seismic cue, with a latency of response (4/7 elephant maximum rate starting 180 s or later).

We also found significant differences in head scanning and freezing behaviour for the treatments containing human-generated noise (WN and Ele&WN; figure 3). Head scanning and freezing sometimes increased during the periods where the seismic tracks were playing: we found increased head scanning and freezing in the treatment versus pretreatment periods (b versus a1, $p = 0.031$ and $p < 0.01$ for WN and Ele&WN head scanning; $p < 0.01$ for Ele&WN freezing; Wilcoxon signed-rank = 27, 36, 36) and increased freezing in the treatment versus post-treatment periods for WN (b versus a2, $p < 0.01$; Wilcoxon signed-rank = 36). Both head scanning and freezing are thought to indicate vigilance or listening in elephants [12,19,22], with freezing also thought to promote seismic wave detection in elephants [7]. Elephants have previously been shown to increase vigilant behaviours in response to seismic and acoustic playback of conspecific calls [12,22].

No other significant differences in behaviours were recorded when the elephants were present, whether trunk swing/manipulation (can be a sign of apprehension [19]) or social and foraging behaviours, although behaviour could not be recorded once the elephant left the site. Compared to acoustic playback of conspecific calls [12,19], there were no differences in 'listening' behaviour (ears extended stiff), which supports that elephants respond differently to seismic and acoustic playback of conspecific calls [12,22].

Retreat response in animals is regarded as defensive, indicating an association between the stimulus and increased risk [10,35]. The retreat response is an output of a trade-off between the advantages of the response (risk-avoidance) versus the costs (less time for beneficial activities such as feeding) [4]. In the case of the sound of bees, the elephants consistently retreat to this acoustic stimulus as they associate it with the risk of stinging [20,32,36,37]. In the case of human voices, elephants are more likely to retreat to Maasai men voices as they associate it with a higher threat [4]. This shows that elephants employ high-level discrimination of acoustic vibrational cues to assess relative risk [4,22], which also appears to be the case with seismic vibrational cues (figures 2 and 3) and signals [22].

The trade-off underlying a retreat response is hypothesized to be similar between predation and non-lethal human disturbance [35], of which seismic noise is one example. Our findings suggest that human-generated noise (e.g. Ele&WN) was associated with a higher risk in elephants. The specific risks encoded within these seismic cues that necessitate a quick retreat response are hard to untangle and warrant further research. It could be that elephants

associate these seismic cues with unusual or unknown human activity, which they could associate with risk [4,11]; i.e. human activities outside of the elephants' experiences could be associated with higher risk (even if the risk is actually low). For example, known/normal human activity associated with low risk could be seismic cues from tourist vehicles during the day for habituated elephants, whereas unusual/unknown human activity might be larger vehicles at night that may be associated with higher risk. This may also explain why there was not a significant response to the elephant-generated treatment—because they were less 'unusual' to the elephants, although the lower seismic amplitude of this treatment could also explain this. Less likely (due to the fast response), it could be that these seismic cues indicate a risk as they are able to mask seismic signals of greater biological importance, or at least reduce the efficacy of necessary communication [18]. Finally, miscategorization of the seismic cues by the elephants cannot be ruled out, where the seismic cues might be misinterpreted as something else elephants associate with risk (e.g. landslide, earthquake). Compared to the sound of bees, the quantitative comparison indicates that the association with risk is not as universal or strongly linked as the risk of bee stings, as fewer individuals showed the retreat response to the combined seismic noise treatment compared to the sound of bees treatment, and the response took longer.

Our results indicate that seismic noise is a cue that matters to elephants in the wild: it is a stimulus type that contains information that elephants responded to, in some cases with risk-avoidance responses. Avoidance behaviour in elephants has also been correlated with an increased presence of stress hormone [38]. Furthermore, seismic noise is often an overlooked form of anthropogenic noise [28,29], so elephant behavioural responses to the seismic noise generated by infrastructure development and use (including roads, railways and human settlements) will be an important avenue for future research. Tracking data suggest that African elephants avoid or change the timing and speed of their movements around newly built roads/railways [5]. Compared to our human-generated treatments, an example recording of car noise was the same order of magnitude for the maximum amplitude levels (*ca* 3E−5 car versus *ca* 1E−5 m/s Ele&WN), was more sustained (*ca* 15% points greater than 1E−5 m/s car versus *ca* 4% WN) and contained similar frequency content (electronic supplementary material, figures S1–S3). This suggests that there will be significant challenges to elephants resulting from their seismic sensitivity as human-generated seismic noise increases in their natural habitats. Overall, we show that the seismic sensory domain contains a wider variety of information for elephants than previously considered, revealing deeper connections between elephants and their dynamic and challenging physical environment.

Ethics. All institutional and national guidelines for the use of protected animals for scientific research were followed during this project. The project was approved through an ethical review process within the University of Oxford by the Zoology Animal Welfare and Ethical Review Board (ref.: APA/1/5/ZOO/NAPSA/Mortimer/Elephant-Vibrations). Research permit and project approval was granted locally from National Commission for Science, Technology and Innovation, Kenya for research in Samburu county (ref.: NACOSTI/P/16/69501/9147). B.M. had project approval and research affiliation from the Kenya Wildlife Service. Park permits were also gathered for Samburu and Buffalo Springs National Reserves. Animal welfare standards were ensured during fieldwork via ongoing evaluation of elephant risk and behaviours during and after experiments. Specifically, the risk of injury to the elephants was sufficiently low to continue planned experiments. All treatments were non-contact in their nature, so caused no physical harm. Treatment periods were kept to 2 min, repeated up to every 6 days for some family/sub-family groups, representing 2/8640 min over this period. The maximum amplitude seismic levels of the chosen treatments were within the natural range that elephants would encounter: they were within the order of magnitude (and slightly lower) than the seismic levels that elephants can generate themselves and those made by cars.

Data accessibility. Electronic supplementary material is provided to support this article [39]. This includes electronic supplementary material, figures S1 and S2 showing geophone and microphone recordings of treatments on different deployments (time–velocity plots and spectrograms respectively); electronic supplementary material, figure S3 showing example geophone recordings of elephant steps and car noise; electronic supplementary material, table showing definitions of behaviours used for video analysis; and electronic supplementary material, movie showing an example of elephant responses to human- and elephant-generated seismic vibrations combined treatment (Ele&WN). The cropped video data (videos separated as a1 silent, b treatment and a2 silent segments: 96 videos) collected and used as part of this manuscript and an explanatory spreadsheet (with video name to treatment key, list of focal individuals and where they are on the video) are available from the Dryad Digital Repository: https://doi.org/10.5061/dryad.3tx95X6gb [40].

Authors' contributions. B.M.: conceptualization, data curation, formal analysis, funding acquisition, investigation, methodology, project administration, resources, validation, visualization, writing-original draft, writing-review and editing; J.A.W.: formal analysis, investigation, methodology, validation, writing-review and editing; D.S.L.: investigation, methodology, validation, writing-review and editing; M.R.: visualization, writing-review and editing; D.D.: methodology, supervision, validation, writing-review and editing.

All authors gave final approval for publication and agreed to be held accountable for the work performed therein.

Competing interests. We declare we have no competing interests.

Funding. B.M. thanks the British Ecological Society (grant no. LRB18/1010), the Royal Commission for the Exhibition of 1851, St Anne's College, Oxford and the Royal Society (grant no. URF/R1/191033) for funding.

Acknowledgements. All authors thank Iain Douglas-Hamilton, George Wittemeyer and all the staff at Save the Elephants for kindly agreeing to support the fieldwork in Kenya. We thank Lucas Wilkins and John Hogg for their help with the modified speaker set-up. B.M. thanks Lucy Taylor and Tom Mulder for their comments on the manuscript. B.M. thanks Fritz Vollrath for his valuable discussions throughout the study. We thank Tarje Nissen-Meyer for the loan of the Raspberry Shake 4D.

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
