## [Peer Review File · Proceedings of the Royal Society B: Biological Sciences]

Review History

RSPB-2021-0774.R0 (Original submission)

Review form: Reviewer 1 (Phyllis Lee)

Recommendation

Accept with minor revision (please list in comments)

Scientific importance: Is the manuscript an original and important contribution to its field?

Good

General interest: Is the paper of sufficient general interest?

Good

Quality of the paper: Is the overall quality of the paper suitable?

Good

Is the length of the paper justified?

Yes

Should the paper be seen by a specialist statistical reviewer?

No

Do you have any concerns about statistical analyses in this paper? If so, please specify them explicitly in your report.

No

It is a condition of publication that authors make their supporting data, code and materials available - either as supplementary material or hosted in an external repository. Please rate, if applicable, the supporting data on the following criteria.

Is it accessible?

Yes

Is it clear?

Yes

Is it adequate?

Yes

Do you have any ethical concerns with this paper?

Yes

Comments to the Author

"Combined track" is unclear in the abstract (combination of what, not yet explained).

18-19 sentence doesn't add information here.

21: how is distance travelled = retreat, unless there is a direction (e.g. away from speaker) associated with distance. Alternatively they could be "interested" and approach. Clarify direction as well as distance.

Reference 1 is a great compendium, but not the original data. Please ensure you actually refer to the relevant authorities rather than a book synthesis. And the evidence for "abiotic" cue recognition may come from the 2005 Nature paper referring to imitation of train vocalisations? Please be clear about what is being detected by elephants (wind noise, "gunshot" large vehicles, helicopters, airplanes, windfarms etc)? This is expanded on in later paragraphs, but the logical flow is a bit lacking at the start.

66-68: if they avoid a seismic domain, then how will they be able to "use it to communicate"?

Unclear phrasing and meaning.

80: the reference [26] looks interesting, but hardly relevant. The question is more likely to be "do elephants respond to human threats perceived from seismic activities (e.g. poachers and gunshot, rangers and helicopters). This is an inappropriate use of the conflict paradigm, unless the meaning is more clearly specified.

103: "their chosen resting locations in the field"? Meaning ambiguous. Located next to rivers or under trees?

104: an iPad has the capacity to capture and then reflect at sufficient power infrasound? Did you ensure that what has played was what you think was heard by elephants? E.g. 106 - how was the seismic domain testing assured. Simply by comparison with no sound?

142: any IOR tests of coding?

143: so no direction was scored and therefore approaching or moving away was not detected.

163: it is clear that "running away from the speaker" is meant when assessing movement. Be more precise.

165-166: if the distance did not differ between trials, then how did they move more in trial with human or high intensity noise? It's a question of how to phrase this more accurately.

211: elephants can differentiate between the average tourist vehicle, research vehicles and large equipment such as lorries or graders. What is the risk associated with in each case? The argument could use a bit more clarification.

224-226: this is too speculative to be useful at this stage. BUT elephant responses to infrastructure development (roads, railways, wind turbines etc) could be highlighted.

229: if the bee noise works, surely that is enough harassment and stress production?

232: ?ever moving? Perhaps "dynamic or challenging"?

You might consider the following refs?

Larom D. and Garstang M., 'Meteorology and elephant infrasound at Etoshsha National Park, Namibia'. Journal of the Acoustics Society of America. Vol 101 Number 3. pp. 1710-1717. 1997

David, A., & Thorne, B. (2013). An underpinning methodology to derive stand-off distances from a wind farm. In 20th International Congress on Sound and Vibration 2013, ICSV 2013.

Review form: Reviewer 2

Recommendation

Major revision is needed (please make suggestions in comments)

Scientific importance: Is the manuscript an original and important contribution to its field?

Excellent

General interest: Is the paper of sufficient general interest?

Excellent

Quality of the paper: Is the overall quality of the paper suitable?

Good

Is the length of the paper justified?

Yes

Should the paper be seen by a specialist statistical reviewer?

No

Do you have any concerns about statistical analyses in this paper? If so, please specify them explicitly in your report.

No

It is a condition of publication that authors make their supporting data, code and materials available - either as supplementary material or hosted in an external repository. Please rate, if applicable, the supporting data on the following criteria.

Is it accessible?

Yes

Is it clear?

Yes

Is it adequate?

Yes

Do you have any ethical concerns with this paper?

Yes

Comments to the Author

Dear editor,

I find the study developed by the authors of high importance and pretty amazing. Furthermore, working with big mammals in remote places increases the challenge, so I congratulate the authors for the experiments.

I have some main questions to raise about the study.

Please feel free to contact me for further questions
The
Kind regards,

Main questions of the work:

The authors talk about noise level, but they don't give values of this or explain in material and methods anything in respect to it. Also, when testing playbacks with different temporal and spectral patterns, it is hard to relate the response to the amplitude, once it could be to the all other parameters of the vibration that diverge between the playbacks. I understand in the third type of playback the amplitude increase, but there is also the combination of the two previous playbacks, therefore you can have the sum of the response to both emissions at the same time. If your question involved the amplitude, I would expect to see the same playback with different levels. Consider not including this as a main question. You have three different playbacks. For me your question is about the elephant, human generated and elephant+human vibrations.

Why choosing white noise to understand the effect of human generated noise instead of a real noise they could encounter, like roads? White noise it is an artificial noise. Even if for comparisons, the study mentioned with acoustic noise tested road noise, so it could have also be compared. Therefore I was curious about the choice.

Material and Methods needs some clarifications.

Line 83 - You talk about levels, where are the levels and the data of it? At least in the Figure, the authors should include the scale. This changes according to setting when building the figure, therefore it is important to know it.

Line 85-86 - how was the playback of the elephants chosen? Was it a sample of mixed animals, was it one animal call, with an average call parameter among the ones found in the place. Could the chosen elephant signal influence the behavior if it was from a different animal? Then why this one?

Line 96-99 - I would suggest to make a figure of the experimental design to make it easier to understand all the stimuli presented. I understood the three playbacks, but the silence and white noises repeated four times is not clear (12 seconds x4). It will be easier to have a diagram of this instead of having to calculate and exchange all the types of stimuli. If you have randomly played all the playbacks to all groups then you should have had a sequence of: (a1, b, a2) + (a1, b, a2) + (a1, b, a2), is that correct? If each of them had 2 minutes then the total time of experiment was 2 x 3 (random playbacks), so 6 min? Where they played in a sequence? So final silence of one was automatically connected to the silence of the other? I had to read it a couple of times to understand it and might even have made a mistake here explaining. The figure of experimental design could easily solve this issue.

Line 96-99 - Why silence for LowEle and LowWN AND white noise for HighEleWN?

Line 132 - I would suggest to include the main behaviors here. The full explanation of them can be in Supplementary material as they are, but it will be important to know which were the behaviors taken into consideration even by reading only the paper.

Figure 1. The color scale should be included. This changes according to setting when building the figure, therefore it is important to know it. Specially once you have combined the two treatments into the playback C, it is interesting to understand the result of it.

How were the authors approaching the elephants with cars and not having any response to it? Did you had any elephants leaving at any time when approaching them with the car? The cars are a real anthropogenic vibrations and very strong. I would expect them to react to it, specially

when seeing the results of the study in which the elephants reacted to all the three playbacks emitted.

If the emission y elephants was recorded with a bull, there should be an alarm call, no? Why they didnt react to it? Why they reacted to white noise and not to the car approaching.

Table 1. what are 1st: 3 ?? this should be mentioned in the legend.

Line 153 – Again, the authors talk about noise level, but they dont give values of this or explain in material and methods anything of this respect.

Fig 2 – indicate colors for each of the treatments.

I missed in the discussion a short part about the other behaviors measured. Even tho they were not significantly different, maybe you were expecting any of them to be different under the noise or maybe any of the other behaviors suffer an effect of acoustic noise in the other studies and you didn't find it with vibrations. Finding differences in the response of the animals to the acoustic and vibrations will be already interesting. I would include a short paragraph in which you briefly discuss the other behaviors and why they might not have been affected.

Decision letter (RSPB-2021-0774.R0)

06-May-2021

Dear Dr Mortimer:

Your manuscript has now been peer reviewed and the reviews have been assessed by two reviewers and an Associate Editor. The reviewers' comments (not including confidential comments to the Editor) and the comments from the Associate Editor are included at the end of this email for your reference. As you will see, the reviewers and the Editors have raised some concerns with your manuscript and we would like to invite you to revise your manuscript to address them. In particular, more detail is needed in the methods section. In addition, it might be interesting to add more discussion about the implication of your results for infrastructure development and how challenging this will be to species that respond to human-produced 24/7 sound. Finally, one of the reviewers raised the issue of a further need to justify the stress caused to the animals (as evidenced by their flight) by the experimental protocol in your ethics statement.

When submitting your revision please upload a file under "Response to Referees" - in the "File Upload" section. This should document, point by point, how you have responded to the reviewers' and Editors' comments, and the adjustments you have made to the manuscript. We

require a copy of the manuscript with revisions made since the previous version marked as 'tracked changes' to be included in the 'response to referees' document.

Research ethics:

Use of animals and field studies:

It is a condition of publication that you make available the data and research materials supporting the results in the article. Please see our Data Sharing Policies (<https://royalsociety.org/journals/authors/author-guidelines/#data>). Datasets should be deposited in an appropriate publicly available repository and details of the associated accession number, link or DOI to the datasets must be included in the Data Accessibility section of the article (<https://royalsociety.org/journals/ethics-policies/data-sharing-mining/>). Reference(s) to datasets should also be included in the reference list of the article with DOIs (where available).

If you wish to submit your data to Dryad (<http://datadryad.org/>) and have not already done so you can submit your data via this link [http://datadryad.org/submit?journalID=RSPB&manu=\(Document not available\)](http://datadryad.org/submit?journalID=RSPB&manu=(Document%20not%20available)), which will take you to your unique entry in the Dryad repository.

Online supplementary material will also carry the title and description provided during submission, so please ensure these are accurate and informative. Note that the Royal Society will not edit or typeset supplementary material and it will be hosted as provided. Please ensure that

the supplementary material includes the paper details (authors, title, journal name, article DOI). Your article DOI will be 10.1098/rspb.[paper ID in form xxxx.xxxx e.g. 10.1098/rspb.2016.0049].

Please submit a copy of your revised paper within three weeks. If we do not hear from you within this time your manuscript will be rejected. If you are unable to meet this deadline please let us know as soon as possible, as we may be able to grant a short extension.

Best wishes,
Dr Sarah Brosnan
Editor, Proceedings B
mailto:proceedingsb@royalsociety.org

Associate Editor
Comments to Author:

Dear authors, as you can see both reviewers had positive things to say about your manuscript, however, both also raised important points and criticisms that would need to be addressed before the manuscript could be accepted for publication. In particular, both referees brought up concerns about the ethics of the study and whether, in light of the behaviour observed in the elephants in response to playback, if the ethical approval of the home university was sufficient. I have also read your manuscript and echo some of the concerns of reviewer 2 about the detail missing from your methods. Specifically, related to the magnitudes of the seismic playbacks, there are values given only in the supplementary figures and then they must be extracted from inspection of the figures. Playback levels in these figures are not clear to me, as the "LowWN" playback appears to be even louder than the "HighEleWN", and both much higher levels than the LowELE. Finally, important for both ethical considerations as well as interpretation of the results would be some information about whether playback levels at the position of the elephants were at, above or below normal levels for what elephants produce themselves or typical automobile vibrations.

Reviewer(s)' Comments to Author:

Referee: 1

Comments to the Author(s)

"Combined track" is unclear in the abstract (combination of what, not yet explained).

18-19 sentence doesn't add information here.

21: how is distance travelled = retreat, unless there is a direction (e.g. away from speaker) associated with distance. Alternatively they could be "interested" and approach. Clarify direction as well as distance.

Reference 1 is a great compendium, but not the original data. Please ensure you actually refer to the relevant authorities rather than a book synthesis. And the evidence for "abiotic" cue recognition may come from the 2005 Nature paper referring to imitation of train vocalisations? Please be clear about what is being detected by elephants (wind noise, "gunshot" large vehicles, helicopters, airplanes, windfarms etc)? This is expanded on in later paragraphs, but the logical flow is a bit lacking at the start.

66-68: if they avoid a seismic domain, then how will they be able to "use it to communicate"?

Unclear phrasing and meaning.

80: the reference [26] looks interesting, but hardly relevant. The question is more likely to be "do elephants respond to human threats perceived from seismic activities (e.g. poachers and gunshot, rangers and helicopters). This is an inappropriate use of the conflict paradigm, unless the meaning is more clearly specified.

103: "their chosen resting locations in the field"? Meaning ambiguous. Located next to rivers or under trees?

104: an iPad has the capacity to capture and then reflect at sufficient power infrasound? Did you ensure that what has played was what you think was heard by elephants? E.g. 106 – how was the seismic domain testing assured. Simply by comparison with no sound?

142: any IOR tests of coding?

143: so no direction was scored and therefore approaching or moving away was not detected.

163: it is clear that “running away from the speaker” is meant when assessing movement. Be more precise.

165-166: if the distance did not differ between trials, then how did they move more in trial with human or high intensity noise? It's a question of how to phrase this more accurately.

211: elephants can differentiate between the average tourist vehicle, research vehicles and large equipment such as lorries or graders. What is the risk associated with in each case? The argument could use a bit more clarification.

224-226: this is too speculative to be useful at this stage. BUT elephant responses to infrastructure development (roads, railways, wind turbines etc) could be highlighted.

229: if the bee noise works, surely that is enough harassment and stress production?

232: ?ever moving? Perhaps “dynamic or challenging”?

You might consider the following refs?

Larom D. and Garstang M., 'Meteorology and elephant infrasound at Etoshsha National Park, Namibia'. *Journal of the Acoustics Society of America*. Vol 101 Number 3. pp. 1710-1717. 1997

David, A., & Thorne, B. (2013). An underpinning methodology to derive stand-off distances from a wind farm. In 20th International Congress on Sound and Vibration 2013, ICSV 2013.

Referee: 2

Comments to the Author(s)

I find the study developed by the authors of high importance and pretty amazing.

Furthermore, working with big mammals in remote places increases the challenge, so I congratulate the authors for the experiments.

I have some main questions to raise about the study.

Main questions of the work:

The authors talk about noise level, but they don't give values of this or explain in material and methods anything in respect to it. Also, when testing playbacks with different temporal and spectral patterns, it is hard to relate the response to the amplitude, once it could be to the all other parameters of the vibration that diverge between the playbacks. I understand in the third type of playback the amplitude increase, but there is also the combination of the two previous playbacks, therefore you can have the sum of the response to both emissions at the same time. If your question involved the amplitude, I would expect to see the same playback with different levels. Consider not including this as a main question. You have three different playbacks. For me your question is about the elephant, human generated and elephant+human vibrations.

Why choosing white noise to understand the effect of human generated noise instead of a real noise they could encounter, like roads? White noise it is an artificial noise. Even if for comparisons, the study mentioned with acoustic noise tested road noise, so it could have also be compared. Therefore I was curious about the choice.

Material and Methods needs some clarifications.

Line 83 - You talk about levels, where are the levels and the data of it? At least in the Figure, the authors should include the scale. This changes according to setting when building the figure, therefore it is important to know it.

Line 85-86 – how was the playback of the elephants chosen? Was it a sample of mixed animals, was it one animal call, with an average call parameter among the ones found in the place. Could the chosen elephant signal influence the behavior if it was from a different animal? Then why this one?

Line 96-99 – I would suggest to make a figure of the experimental design to make it easier to understand all the stimuli presented. I understood the three playbacks, but the silence and white noises repeated four times is not clear (12 seconds x4). It will be easier to have a diagram of this instead of having to calculate and exchange all the types of stimuli. If you have randomly played all the playbacks to all groups then you should have had a sequence of: (a1, b, a2) + (a1, b, a2) + (a1, b, a2), is that correct? If each of them had 2 minutes then the total time of experiment was 2 x 3 (random playbacks), so 6 min? Where they played in a sequence? So final silence of one was automatically connected to the silence of the other? I had to read it a couple of times to understand it and might even have made a mistake here explaining. The figure of experimental design could easily solve this issue.

Line 96-99 - Why silence for LowEle and LowWN AND white noise for HighEleWN?

Line 132 – I would suggest to include the main behaviors here. The full explanation of them can be in Supplementary material as they are, but it will be important to know which were the behaviors taken into consideration even by reading only the paper.

Figure 1. The color scale should be included. This changes according to setting when building the figure, therefore it is important to know it. Specially once you have combined the two treatments into the playback C, it is interesting to understand the result of it.

How were the authors approaching the elephants with cars and not having any response to it? Did you had any elephants leaving at any time when approaching them with the car? The cars are a real anthropogenic vibrations and very strong. I would expect them to react to it, specially when seeing the results of the study in which the elephants reacted to all the three playbacks emitted.

If the emission y elephants was recorded with a bull, there should be an alarm call, no? Why they didnt react to it? Why they reacted to white noise and not to the car approaching.

Table 1. what are 1st: 3 ?? this should be mentioned in the legend.

Line 153 – Again, the authors talk about noise level, but they dont give values of this or explain in material and methods anything of this respect.

Fig 2 – indicate colors for each of the treatments.

I missed in the discussion a short part about the other behaviors measured. Even tho they were not significantly different, maybe you were expecting any of them to be different under the noise or maybe any of the other behaviors suffer an effect of acoustic noise in the other studies and you didn't find it with vibrations. Finding differences in the response of the animals to the acoustic and vibrations will be already interesting. I would include a short paragraph in which you briefly discuss the other behaviors and why they might not have been affected.

Author's Response to Decision Letter for (RSPB-2021-0774.R0)

See Appendix A.

Decision letter (RSPB-2021-0774.R1)

08-Jun-2021

Dear Dr Mortimer

I am pleased to inform you that your manuscript entitled "Noise matters: Elephants show risk-avoidance behaviour in response to human-generated seismic cues" has been accepted for publication in Proceedings B.

Data Accessibility section

Open Access

Your article has been estimated as being 8 pages long. Our Production Office will be able to confirm the exact length at proof stage.

Paper charges

Sincerely,

Dr Sarah Brosnan

Associate Editor:

Comments to Author:

Dear Authors,

Thank you for the interesting and well-presented manuscript. I believe the revision has addressed the key concerns the reviewers and I have raised.

Appendix A

Editor

Your manuscript has now been peer reviewed and the reviews have been assessed by two reviewers and an Associate Editor. The reviewers' comments (not including confidential comments to the Editor) and the comments from the Associate Editor are included at the end of this email for your reference. As you will see, the reviewers and the Editors have raised some concerns with your manuscript and we would like to invite you to revise your manuscript to address them. In particular, more detail is needed in the methods section. In addition, it might be interesting to add more discussion about the implication of your results for infrastructure development and how challenging this will be to species that respond to human-produced 24/7 sound. Finally, one of the reviewers raised the issue of a further need to justify the stress caused to the animals (as evidenced by their flight) by the experimental protocol in your ethics statement.

Thank you for your helpful comments. As outlined below, we have now addressed the Editor's and reviewer's concerns regarding the methods section, and have added the extra detail requested (see Methods and Supplementary Methods) and revised Figures 1 and 2, Supplementary Figure 2 and provided a new Supplementary Figure 3. We have added some discussion regarding infrastructure development to the introduction (lines 59, 75-81) and discussion (256-263), including the challenges that anthropogenic noise poses to elephants (263-265). We have also expanded the methods section (lines 117-125) and the Ethics statement around the ethical issues relevant here to justify our experimental procedure.

Associate Editor

Comments to Author:

Dear authors, as you can see both reviewers had positive things to say about your manuscript, however, both also raised important points and criticisms that would need to be addressed before the manuscript could be accepted for publication. In particular, both referees brought up concerns about the ethics of the study and whether, in light of the behaviour observed in the elephants in response to playback, if the ethical approval of the home university was sufficient.

Thank you for your helpful comments and suggestions – we think the paper is much improved and clearer as a result of this revision. To address the first point, project approval was obtained at the local level (as well as the home university) via a research permit with NACOSTI and a research affiliation with KWS. Part of the ethical approval from the home university explicitly stated that experiments should be stopped if risk of injury was observed. In line with this requirement, we monitored the risk of injury to elephants during experiments. Even with the running response, the risk was deemed to be low as elephants were in control of their movements (see video data accompanying manuscript). Furthermore, the treatment was extremely limited in duration and infrequent for each group – the treatment period was only 2 minutes, with repeat treatments after a minimum of 6 days for each family/sub-family group. Also relevant is that the stimulus is non-contact without any physical harm. These points have been clarified in the methods section (lines 117-125, 137-138) and Ethics statement accompanying the manuscript.

I have also read your manuscript and echo some of the concerns of reviewer 2 about the detail missing from your methods. Specifically, related to the magnitudes of the seismic playbacks, there are values given only in the supplementary figures and then they must be extracted from inspection of the figures. Playback levels in these figures are not clear to me, as the "LowWN" playback appears to be even louder than the "HighEleWN", and both much higher levels than the LowELE.

We are sorry for this omission - levels are now included in spectrograms in Figure 1 and Supplementary Figure 2 and the new Supplementary Figure 3. We have also added the absolute maximum magnitudes of seismic recordings at 3, 10 and 30 m from playback location of each of the tracks and example background noise in the Methods section (lines 109-115). In response to a comment from reviewer 2, we have also de-emphasised the

focus on 'levels' of noise, concentrating instead on the type of noise (human/elephant generated), with text, figures, tables and movie edited throughout to reflect this.

Finally, important for both ethical considerations as well as interpretation of the results would be some information about whether playback levels at the position of the elephants were at, above or below normal levels for what elephants produce themselves or typical automobile vibrations.

The playback levels, quantified here as maximum amplitude, at c.17 m (average distance between playback equipment and focal elephant) would be below $1E-5$ m/s. This is below the range that we recorded for some elephant steps (c. $3E-5$ m/s) and car noise (c. $4E-5$ m/s) at c.20m from source, with the same recording equipment and similar terrain. These example maximum amplitudes have now been added to the Methods section (lines 133-135), included in a new Supplementary Figure 3 and video data related to these recordings has been uploaded to Dryad. Car noise is referred to in the discussion (lines 260-263). The levels are also referred to in the revised Ethics Statement. We have included more detail on this comparison (caveats of what affects amplitude and what factors other than amplitude will be important) in the Supplementary Methods.

We hope the changes we have made in response to your comments meet your satisfaction. Please see responses to the reviewers' comments below.

Reviewer(s)' Comments to Author:

Referee: 1

Comments to the Author(s)

"Combined track" is unclear in the abstract (combination of what, not yet explained).

This now reads "combined track (elephant- and human-generated)." line 18

18-19 sentence doesn't add information here.

This has now been removed.

21: how is distance travelled = retreat, unless there is a direction (e.g. away from speaker) associated with distance. Alternatively they could be "interested" and approach. Clarify direction as well as distance.

Yes, distance was away from the trial location. This has been clarified, lines 21-22, 153, 157, 159-161, 166-167, 177-179, 183-184.

Reference 1 is a great compendium, but not the original data. Please ensure you actually refer to the relevant authorities rather than a book synthesis. And the evidence for "abiotic" cue recognition may come from the 2005 Nature paper referring to imitation of train vocalisations? Please be clear about what is being detected by elephants (wind noise, "gunshot" large vehicles, helicopters, airplanes, windfarms etc)? This is expanded on in later paragraphs, but the logical flow is a bit lacking at the start.

Reference 1 has now been removed and in line with this suggestion, we have added four specific primary literature references (references 2,3,5,6). We have been more specific on the evidence for each type of signal/cue (e.g. human) and what is being detected by elephants (e.g. presence of roads/railway) in this sentence. To avoid confusion, we have replaced 'abiotic' with 'natural earth processes', as this is what we meant by this term, e.g. lightning, rainfall etc (rather than human-generated cues) and we now have a specific reference for this (Garstang et al. 2014). Lines 28-30.

66-68: if they avoid a seismic domain, then how will they be able to "use it to communicate"? Unclear phrasing and meaning.

This has now been rephrased for clarity. Lines 66-68.

80: the reference [26] looks interesting, but hardly relevant. The question is more likely to be “do elephants respond to human threats perceived from seismic activities (e.g. poachers and gunshot, rangers and helicopters). This is an inappropriate use of the conflict paradigm, unless the meaning is more clearly specified.

Our meaning was more to do with impacts of anthropogenic seismic noise on elephants rather than human threats specifically and we have now clarified this by referring to infrastructure development. Reference to conflict has been removed and we have edited to make the relevance of old reference 26 (now 31) more explicit. We have added two additional references (26, 32) to support that infrastructure development will increase seismic noise. Lines 78-81.

103: “their chosen resting locations in the field”? Meaning ambiguous. Located next to rivers or under trees?

This has now been clarified as under trees. Line 101.

104: an iPad has the capacity to capture and then reflect at sufficient power infrasound? Did you ensure that what has played was what you think was heard by elephants? E.g. 106 – how was the seismic domain testing assured. Simply by comparison with no sound?

We also used an amplifier (so iPad - > amplifier - > speaker) and have added this detail lines 102-103. We trialled and recorded our playback tracks at various distances from the speaker (3, 10 and 30 m) to check the levels following coupling with the ground and propagation to the distance at which the elephants were on average (c. 17m). This has been clarified lines 104-105, 108-113, 132-133. Testing was assured in the seismic domain by comparison to background seismic noise (i.e. no treatment), which is shown in Supplementary Figures 2 and 3 and data is now quoted lines 111-115.

142: any IOR tests of coding?

Not in this case as coding only differed in 3/252 photo pairs. As coding was binary (yes/no), with repeated measures (3 treatments, 12 time periods, and 7 families), any statistical test for whether the coding was different between the two people would have been complex or so averaged that it rendered the test meaningless. This has been clarified in the Supplementary Methods. The videos are available on Dryad.

143: so no direction was scored and therefore approaching or moving away was not detected.

It is correct that direction was not scored from the data in King et al. 2007, but from this reference and King et al. 2010, it is clear that elephants moved away from the source of the acoustic sound of bees. This has been clarified lines 166-167.

163: it is clear that “running away from the speaker” is meant when assessing movement. Be more precise.

Yes, running away from source location. This has been clarified to be more precise lines 21-22, 153, 157, 159-161, 166-167, 177-179, 183-184.

165-166: if the distance did not differ between trials, then how did they move more in trial with human or high intensity noise? It’s a question of how to phrase this more accurately.

We are sorry for the lack of clarity here. This has now been rephrased to be more accurate lines 186-188.

211: elephants can differentiate between the average tourist vehicle, research vehicles and large equipment such as lorries or graders. What is the risk associated with in each case? The argument could use a bit more clarification.

Our argument was more to do with ‘unusual’ human activity causing this response, i.e. stimuli not encountered before but associated with humans, rather than a learned response where they associate the stimulus (e.g. seismic noise from different vehicles) with risk. This argument has been clarified lines 237-242.

224-226: this is too speculative to be useful at this stage. BUT elephant responses to infrastructure development (roads, railways, wind turbines etc) could be highlighted.

This section has been rephrased to focus on infrastructure development, as suggested, lines 256-265.

229: if the bee noise works, surely that is enough harassment and stress production?

This has now been removed.

232: ?ever moving? Perhaps “dynamic or challenging”?

We have changed this to ‘dynamic and challenging’ line 267.

You might consider the following refs?

Larom D. and Garstang M., ‘Meteorology and elephant infrasound at Etoshya National Park, Namibia’. *Journal of the Acoustics Society of America*. Vol 101 Number 3. pp. 1710-1717. 1997

David, A., & Thorne, B. (2013). An underpinning methodology to derive stand-off distances from a wind farm. In 20th International Congress on Sound and Vibration 2013, ICSV 2013.

Thank you for these suggestions, we have added the first reference (reference 17) and have chosen an alternative reference for seismic noise generated by wind turbines (reference 26). We hope we have addressed your concerns with the changes we have made to the manuscript.

Referee: 2

Comments to the Author(s)

I find the study developed by the authors of high importance and pretty amazing. Furthermore, working with big mammals in remote places increases the challenge, so I congratulate the authors for the experiments.

We thank the reviewer for their positive comments and have made edits to address the comments you have raised.

I have some main questions to raise about the study.

Main questions of the work:

The authors talk about noise level, but they don't give values of this or explain in material and methods anything in respect to it. Also, when testing playbacks with different temporal and spectral patterns, it is hard to relate the response to the amplitude, once it could be to the all other parameters of the vibration that diverge between the playbacks. I understand in the third type of playback the amplitude increase, but there is also the combination of the two previous playbacks, therefore you can have the sum of the response to both emissions at the same time. If your question involved the amplitude, I would expect to see the same playback with different levels. Consider not including this as a main question. You have three different playbacks. For me your question is about the elephant, human generated and elephant+human vibrations.

We thank the reviewer for their helpful suggestion and we agree that rephrasing the treatments as elephant, human or elephant+human generated is a better and simpler framework for the study. We have made changes throughout the abstract, manuscript, figures, tables, supplementary material and movie to rename the seismic playback treatments and focus on their different types (elephant-generated, human-generated or both). We have also provided more information on the levels involved in Figure 1, Supplementary Figure 2, new Supplementary Figure 3 and in the methods lines 108-115.

Why choosing white noise to understand the effect of human generated noise instead of a real noise they could encounter, like roads? White noise it is an artificial noise. Even if for comparisons, the study mentioned with acoustic noise tested road noise, so it could have also be compared. Therefore I was curious about the choice.

White noise was chosen as it has previously been used in other acoustic playback studies (e.g. King et al. 2007), so provided an interesting comparison between acoustic and seismic domain. We agree that using recordings of road noise would be an excellent avenue for future studies. The choice of white noise is now mentioned in the Supplementary methods.

Material and Methods needs some clarifications.

Line 83 - You talk about levels, where are the levels and the data of it? At least in the Figure, the authors should include the scale. This changes according to setting when building the figure, therefore it is important to know it.

The scale for amplitude is now included in the figures – Figure 1 (Volts, as these are the input tracks), Supplementary Figure 2 (m/s for geophone, V for microphone) and new Supplementary Figure 3 (m/s for geophone). The maximum amplitude levels of geophone recordings are also given in the methods (lines 111-115) and discussion (lines 262-263).

Line 85-86 – how was the playback of the elephants chosen? Was it a sample of mixed animals, was it one animal call, with an average call parameter among the ones found in the place. Could the chosen elephant signal influence the behavior if it was from a different animal? Then why this one?

We had some source functions from our previous study (reference 15), and picked a single recording where there were no visible elephant rumbles/calls in the spectrograms (see Supplementary Figure 2), but lots of elephant activity. Our reasoning was that locomotion-generated cues might have less social information compared to an elephant call, so we would predict that cues from different animals would be similar. This was important to make sure that familiarity with the sender of the seismic vibration was not a confounding variable. Due to time constraints we unfortunately did not have time to generate more source functions, for example rumbles from different elephants, so were limited to the data we had already collected. We agree that studies investigating how the identity of the elephant and the social and behavioural context affects the seismic vibrations that elephants generate would be an interesting avenue for further study and future seismic playback experiments. However data on this is currently lacking. The choice of the elephant-generated track has been clarified in the Supplementary Methods.

Line 96-99 – I would suggest to make a figure of the experimental design to make it easier to understand all the stimuli presented. I understood the three playbacks, but the silence and white noises repeated four times is not clear (12 seconds x4). It will be easier to have a diagram of this instead of having to calculate and exchange all the types of stimuli. If you have randomly played all the playbacks to all groups then you should have had a sequence of: (a1, b, a2) + (a1, b, a2) + (a1, b, a2), is that correct? If each of them had 2 minutes then the total time of experiment was 2 x 3 (random playbacks), so 6 min? Where they played in a sequence? So final silence of one was automatically connected to the silence of the other? I had to read it a couple of times to understand it and might even have made a mistake here explaining. The figure of experimental design could easily solve this issue.

We are sorry for this lack of clarity here. The above is nearly correct, but each track was only 6 minutes long and one treatment type was played each trial, separating any repeat treatments for each family/sub-family group by at least 6 days. As suggested, we have added a panel to Figure 1 to better communicate the design of the playback treatments. We have clarified our language around ‘tracks’ and ‘treatments’ and edited the methods to make the experimental design clearer, lines 84-98, 127-128, 137-138.

Line 96-99 - Why silence for LowEle and LowWN AND white noise for HighEleWN?

This makes the Ele & WN treatment similar to the Ele track but with superimposed noise on top. This would be similar to having the same track in a high seismic noise environment (e.g. due to human-generated seismic noise). Ele and WN then both had silence so they differed only in the 18 s segments. The choice of silence and WN has been clarified in the Supplementary Methods.

Line 132 – I would suggest to include the main behaviors here. The full explanation of them can be in Supplementary material as they are, but it will be important to know which were the behaviors taken into consideration even by reading only the paper.

This has now been included line 150.

Figure 1. The color scale should be included. This changes according to setting when building the figure, therefore it is important to know it. Specially once you have combined the two treatments into the playback C, it is interesting to understand the result of it.

The scale for amplitude is now included in the figures – Figure 1 (Volts, as these are the input tracks), Supplementary Figure 2 (m/s for geophone, V for microphone) and new Supplementary Figure 3 (m/s for geophone).

How were the authors approaching the elephants with cars and not having any response to it? Did you had any elephants leaving at any time when approaching them with the car? The cars are a real anthropogenic vibrations and very strong. I would expect them to react to it, specially when seeing the results of the study in which the elephants reacted to all the three playbacks emitted.

This particular population of elephants at Samburu are habituated to research vehicles and tourist cars. As such they were a great population with which to do these experiments as we could get within 30 m of them for playback experiments whilst they were resting in the middle of the day. We only conducted experiments if the elephant remained in their location long enough for the playback treatment to start (i.e. time to set up and all of a 1 control period time). A few groups did chose to leave before playback could start, but they were in the minority for our experiments. We think this is because elephants are classifying ‘unusual’ human-generated vibrations from ‘known’/‘usual’ ones, which would include car seismic vibrations. They would therefore associate ‘unusual’ human activity with risk, even if the risk level is not known – a ‘better be safe than sorry’ strategy regarding humans. We might therefore predict that elephants that are less habituated to human activities, such as cars, might show retreat responses to more types of human-generated noise. This argument has been clarified lines 237-242. The experience of the elephant population with cars has been clarified in the Supplementary Methods.

If the emission y elephants was recorded with a bull, there should be an alarm call, no?

We saw no evidence of a rumble in this source function, see spectrogram Supplementary Figure 2a.

Why they didnt react to it? Why they reacted to white noise and not to the car approaching.

We think this is either related to the lower amplitude of this track compared to the other two, or that it was less ‘unusual’ compared to the human-generated tracks. This has been expanded upon lines 242-244.

Table 1. what are 1st: 3 ?? this should be mentioned in the legend.

This is now clearer in Table 1 legend.

Line 153 – Again, the authors talk about noise level, but they dont give values of this or explain in material and methods anything of this respect.

We now do not refer to levels specifically when comparing the treatments – more whether human or elephant-generated The scale for amplitude is now included in the figures – Figure 1 (Volts, as these are the input tracks), Supplementary Figure 2 (m/s for geophone, V for microphone) and new Supplementary Figure 3 (m/s for geophone). The maximum amplitude levels of geophone recordings are also given in the methods (lines 111-115) and discussion (lines 262-263).

Fig 2 – indicate colors for each of the treatments.

This is now clearer on the Figure and in the legend for Figure 2.

I missed in the discussion a short part about the other behaviors measured. Even tho they were not significantly different, maybe you were expecting any of them to be different under the noise or maybe any of the other behaviors suffer an effect of acoustic noise in the other studies and you didn't find it with vibrations. Finding differences in the response of the animals to the acoustic and vibrations will be already interesting. I would include a short paragraph in which you briefly discuss the other behaviors and why they might not have been affected.

We have now added a short paragraph to the discussion to discuss the other behaviours and make a comparison between acoustic and seismic playback lines 219-224.